# Intercomparison and validation of first GLORIA-B measurements of stratospheric and upper tropospheric long-lived tracers and photochemically active species

Gerald Wetzel<sup>1</sup>, Anne Kleinert<sup>1</sup>, Sören Johansson<sup>1</sup>, Felix Friedl-Vallon<sup>1</sup>, Michael Höpfner<sup>1</sup>, Jörn Ungermann<sup>2</sup>, Tom Neubert<sup>3</sup>, Valéry Catoire<sup>4</sup>, Cyril Crevoisier<sup>5</sup>, Andreas Engel<sup>6</sup>, Thomas Gulde<sup>1</sup>, Patrick Jacquet<sup>4</sup>, Oliver Kirner<sup>7</sup>, Erik Kretschmer<sup>1</sup>, Thomas Kulessa<sup>3</sup>, Johannes C. Laube<sup>2</sup>, Guido Maucher<sup>1</sup>, Hans Nordmeyer<sup>1</sup>, Christof Piesch<sup>1</sup>, Peter Preusse<sup>2</sup>, Markus Retzlaff<sup>2</sup>, Georg Schardt<sup>3</sup>, Johan Schillings<sup>3</sup>, Herbert Schneider<sup>3</sup>, Axel Schönfeld<sup>2</sup>, Tanja Schuck<sup>6</sup>, Wolfgang Woiwode<sup>1</sup>, Martin Riese<sup>2,8</sup>, and Peter Braesicke<sup>1,\*</sup>

<sup>1</sup>Institute of Meteorology and Climate Research Atmospheric Trace Gases and Remote Sensing (IMKASF), Karlsruhe Institute of Technology, Karlsruhe, Germany

<sup>2</sup>Institute of Climate and Energy Systems - Stratosphere (ICE-4), Forschungszentrum Jülich, Jülich, Germany <sup>3</sup>Institute of Technology and Engineering (ITE), Forschungszentrum Jülich, Jülich, Germany

<sup>4</sup>Laboratoire de Physique et Chimie de l'Environnement et de l'Espace (LPC2E/CNRS), Université Orléans, Orléans, France

<sup>5</sup>Laboratoire de Météorologie Dynamique, IPSL, CNRS, Palaiseau Cedex, France

<sup>6</sup>Institute for Atmospheric and Environmental Sciences, Goethe Universität, Frankfurt, Germany

<sup>7</sup>Scientific Computing Center (SCC), Karlsruhe Institute of Technology, Karlsruhe, Germany

<sup>8</sup>Institute for Atmospheric and Environmental Research, University of Wuppertal, Wuppertal, Germany

<sup>\*</sup>now at: Business Area Research and Development, Deutscher Wetterdienst, Offenbach am Main, Germany

Correspondence to: Gerald Wetzel (gerald.wetzel@kit.edu)

#### **Abstract**





Accurate observations of the vertical distribution and variability of atmospheric trace gases are essential for understanding chemical processes, validating atmospheric models, and monitoring the impact of anthropogenic emissions on climate and ozone. The Gimballed Limb Observer for Radiance Imaging of the Atmosphere (GLORIA) is a limb-imaging Fourier-Transform Spectrometer (iFTS) designed to provide high-resolution mid-infrared spectra in the 780-1400 cm<sup>-1</sup> wavenumber range. Originally developed for aircraft, the instrument has now been adapted for stratospheric balloon deployment (GLORIA-B) to extend its observational range from the middle troposphere to the middle stratosphere. GLORIA-B completed its first flight from Kiruna (Sweden) in August 2021 and a second from Timmins (Canada) in August 2022 as part of the EU Research Infrastructure HEMERA (Integrated access to balloon-borne platforms for innovative research and technology). The main objectives of these flights were technical

qualification and the provision of a first imaging hyperspectral limb-emission dataset from 5 to 36 km altitude. Here, we present a characterization and validation of GLORIA-B performance using vertical volume mixing ratio (VMR) profiles retrieved from the August 2021 flight. Comparisons with in-situ measurements (ozonesonde, MegaAirCore, and cryosampler) show agreement within 10 % for  $O_3$ ,  $CH_4$ ,  $SF_6$ , and CFC-12, and within 10-20 % for CFC-11, HCFC-22, and CFC-113 up to 18 km, with larger deviations above this altitude. Another objective is analyzing diurnal changes in photochemically active species ( $N_2O_5$ ,  $NO_2$ ,  $CIONO_2$ ,  $BrONO_2$ ). Observed VMR variations align well with simulations from the EMAC (ECHAM5/MESSy Atmospheric Chemistry) chemistry-climate model, though absolute concentrations differ to a certain extent. Nighttime  $BrONO_2$  measurements allowed an estimate of lower stratospheric  $Br_y$  (20.4  $\pm$  2.5 pptv). These results demonstrate the suitability of balloon-borne limb-imaging spectroscopy for providing high-quality vertical trace gas profiles, offering valuable new data to improve our understanding of stratospheric composition and to support the validation of atmospheric models.





40

45

#### 1 Introduction

Balloon-borne mid-infrared limb-emission remote sensing measurements are an essential tool to monitor the temporal and spatial distributions of many trace gases in the stratosphere and upper troposphere with high vertical resolution below the flight altitude (Fischer et al., 2008). Limb-emission spectrometers have the great advantage with respect to solar occultation instruments that they can be used in a flexible way during day and night conditions, being independent of any light source (Friedl-Vallon et al., 2004). However, cooled spectrometers demand more technical effort compared to uncooled devices. This is a reason why the stratospheric balloon missions of such instruments were always limited to a small number (Carli et al., 1984; Murcray et al., 1984; Brasunas et al., 1988; Johnson et al., 1995). For instance, Michelson Interferometer for Passive Atmospheric Sounding (MIPAS) instruments were developed at the Karlsruhe Institute of Technology (KIT). Beside an aircraft instrument, two balloon versions of MIPAS (Fischer and Oelhaf, 1996; Friedl-Vallon et al., 2004) participated in many scientific campaigns between 1989 and 2014 (e.g. von Clarmann et al., 1993; Oelhaf et al., 1994; Wetzel et al., 1995; von Clarmann et al., 1997; Wetzel et al., 1997; Stowasser et al., 1999; Waibel et al., 1999; Höpfner et al., 2002; Stowasser et al., 2002, 2003; Wetzel et al., 2002; Wiegele et al., 2009; Wetzel et al., 2010, 2012, 2015, 2017). In addition, these MIPAS instruments were precursors of the MIPAS Envisat satellite mission (Fischer et al., 2008) and important components for the validation of MIPAS Envisat (e.g. Wetzel et al., 2022) and other satellite instruments like the Global Ozone Monitoring by Occultation of Stars (GOMOS; Renard et al., 2008), the second Improved Limb Atmospheric Spectrometer (ILAS-II; Wetzel et al., 2006), and the Superconducting Submillimeter-Wave Limb-Emission Sounder (SMILES; Sagawa et al., 2013).

While limb-scanning spectrometers like MIPAS need some time (about one minute) to measure a full vertical trace gas profile, the new generation of imaging spectrometers capture a vertical profile at once and even with enhanced vertical resolution. This advantage led to the first aircraft-borne development of the Gimballed Limb Observer for Radiance Imaging of the Atmosphere (GLORIA; Friedl-Vallon et al., 2014; Riese et al., 2014) realized in a collaboration between KIT and the Forschungszentrum Jülich (FZJ). In order to enhance the vertical range of GLORIA to observations in the middle stratosphere albeit still reaching down to the middle troposphere, the instrument was adapted to measure from a stratospheric balloon platform (called GLORIA-B). The first flight was performed from Kiruna (northern Sweden) in August 2021.

Here we show the technical qualification and the provision of a first imaging hyperspectral limb-emission dataset from 5 to 36 km altitude. We present examples of selected retrieval results including uncertainty estimates, altitude resolution as well as long-lived tracer volume mixing ratio (VMR) comparisons to external datasets. In addition, diurnal variations of photochemically active gases are compared to simulations of the chemistry-climate model ECHAM5/MESSy Atmospheric Chemistry (EMAC; Jöckel et al., 2010).

90

95





#### 2 GLORIA-B instrument, data analysis and modelling

In the following subsections, we give an overview of the GLORIA balloon instrument, together with the corresponding data analysis and a description of the atmospheric modelling performed for this study.

# 2.1 GLORIA-B instrument

GLORIA-B is a modification of the airborne instrument (GLORIA-AB). It is a cryogenic limb-imaging Fourier-Transform Spectrometer (iFTS) operating in the thermal infrared spectral

region in the wavelength range between 7 µm and 13 µm using a two-dimensional detector array. It observes 128 vertical and 48 horizontal interferograms per measurement. The instrument provides two-sided interferograms with a maximum optical path difference (MOPD) of up to 8 cm, corresponding to a spectral sampling of 0.0625 cm<sup>-1</sup> (Friedl-Vallon et al., 2014; Riese et al., 2014). The balloon borne version of GLORIA has been developed using the airborne version as basis. The balloon application calls for some fundamental modifications in the design of the instrument. Image rotation by roll movements of the carrier is small and the azimuth viewing direction can be controlled by the gondola, such that only the elevation needs to be steered and stabilised by the instrument. This allows for significant simplification by a replacement of the gimballed frame with a frame with an elevation axis only. On the other hand, the weight of the instrument should be as low as possible for the balloon application. Power consumption is not an issue on an aircraft but batteries on a balloon provide only a limited amount of power. At the high float altitude of a stratospheric balloon with pressure levels of only a few hPa, convective cooling does not work anymore, therefore the thermal management has to be performed by radiation only. The electronic system must be modified to take these requirements and the different temperature and pressure ranges into account. The major modifications with respect to the aircraft instrument are outlined in the following.

#### 2.1.1 Mechanical modifications







The following mechanical modifications have been implemented to integrate the GLORIA spectrometer in the balloon gondola:

- The function of the outer frames of the gimbal frame is taken by the steering of the gondola. The two outer frames, azimuth frame and the image rotation frame, are thus omitted. The design of the innermost frame, the elevation frame, including bearings and direct drive, has been used in the new carrier for GLORIA-B.
- The instrument mounting structure has been manufactured almost completely from aluminium and optimised regarding weight.
- The junction between the mounting structure and the gondola is realised by six steel wire rope isolators.
- The Inertial Measurement and Control Unit (IMCU) is thermally insulated from the mounting structure by a ring made from glass reinforced plastic.
- The blackbody system has been completely redesigned (see section 2.1.5).

During the design process, iterative calculations with finite element analysis provided the optimal solution regarding material utilisation, required rigidity, and sufficient stability. This approach ensures compliance with both the scientific specification regarding the stability of the line of sight and the safety requirements imposed by the balloon operator (CNES).



The final design results in a total mass of 128 kg, roughly half of the mass of the GLORIA-AB, with the centre of gravity centrally located within the structure. The dimensions in length x width x height are as follows: 1290 mm x 1200 mm x 580 mm. Figure 1 shows the carrier with all its components mounted. The instrument integrated into the balloon gondola is displayed in Fig. 2.

**Figure 1.** GLORIA-B instrument carrier with different mounted components as seen from top (a) and bottom (b).

Figure 2. GLORIA-B instrument after integration into the balloon gondola.

# 145 2.1.2 Modifications of the pointing system


The new elevation frame allows the turning of the viewing axis within an angular range from -105° to +75°. It allows pitch stabilization and pointing to different targets: limb, deep space, calibration blackbody, and nadir. In the park position at -40° the interferometer's view is directed into the black body, enabling calibration. The azimuth of the whole gondola is directed and stabilized by an independent azimuth control system operated by a dedicated Centre National d'Études Spatiales (CNES) team. The amplitude of the oscillations in roll and pitch on float height amount only to some tenths of a degree. Therefore, the roll angle is not

stabilized, however measured by the navigation system, such that it can be considered in the retrieval. The software for the IMCU was adapted to this single axis task by the developer iMAR Navigation GmbH, Germany. A new additional communication interface was implemented to have the possibility for direct control of the azimuth stabilization system of CNES by GLORIA-B for special measurement options.

# 2.1.3 Modifications of the thermal concept







In contrast to the situation on the aircraft, there is no convective heat transfer at balloon float altitude (35 to 40 km), therefore only radiative cooling can be utilized. During ascent, however, the instrument is exposed to strong air flow, and care has to be taken that the electronics do not become too cold.

In order to design a thermal concept for the instrument, a ray tracing model has been used to calculate, which part of the surfaces of the components is exposed to space and which part is exposed to Earth or other parts of the gondola. Based on these results, thermal fluxes have been modelled and thermal coupling, insulation, radiative surfaces and reflection shields have been designed such that electronics and instrument stay within their respective operation ranges during ascent and float.

The spectrometer together with the elevation electronics and the interferometer electronics is mounted on the elevation plate, which serves as heat sink. The spectrometer housing itself is painted with white Nextel Suede Coating 3101 428-04, reflecting in the visible and emitting in the infrared, such that radiative cooling is possible whereas sun radiation onto the instrument does not lead to too much heating. The elevation electronics and the IMCU are equipped with an insulation cover to avoid too much cooling during ascent. During float, heat from these components is conducted to the elevation plate.

In the aircraft instrument, detector and compressor are cooled with fans. In the balloon configuration, two heatpipes have been mounted to the detector finger and the compressor, respectively, which are connected to the elevation plate.

The onboard computer, which is also painted in white, has been mounted onto an extra plate to take up the heat. Both the plate and the onboard computer itself are exposed to space for radiative cooling. A reflective plate is mounted next to the spectrometer to allow for further radiation into space.

# 2.1.4 Electrical design







The electrical design of the GLORIA instrument is based on a distributed embedded system architecture, where individual components are coordinated and controlled via a communication network. By integrating autonomous control units within each functional module, significant advantages in terms of scalability, fault tolerance and real-time capability have been achieved.

A key feature of this architecture is the decentralized distribution of the computational load, where multiple embedded control units independently process local operations, thereby relieving the on-board computer. This contributes to increased system reliability, as the failure of individual components does not necessarily affect the entire system. In addition, the modular structure allows efficient maintenance and expansion: defective modules can be selectively replaced and new functionalities can be integrated into the existing system without extensive modifications.

In adapting GLORIA for use as a balloon instrument, a significant proportion of the existing system components could be retained unchanged. Essential modules such as the detector electronics, the visual camera, the interferometer electronics in the spectrometer, the pointing control unit (IMCU) and the direct control of the elevation drive remained largely in their original design. Only the firmware of these components was reconfigured to meet the changed operational requirements.

In order to successfully integrate GLORIA-B into the balloon gondola, it was necessary to make specific modifications to the electrical interfaces and to optimize the power consumption in order to efficiently use the limited battery capacity. As part of these adaptations, a new compact Power Distribution Unit (PDU) with extensive housekeeping data acquisition was developed, specifically tailored to the remaining sub-components of GLORIA-B. The implementation of highly efficient, isolated voltage converters and an integrated power sequencing system for individual functional components enabled intelligent power management. This ensures that power consumption remains largely constant across different measurement configurations, avoiding additional peak loads on the batteries. The high energy efficiency of up to 90% of the new customized PDU has led to a significant reduction in the average total power consumption to 320 watts for the remaining sub-components. This represents a reduction of more than a factor of 3 compared to the original aircraft configuration.

Another modification concerns the on-board computer, which was previously located in the passenger cabin of the aircraft and has now been integrated directly into the instrument carrier (Neubert et al., 2021). The computing unit and the uninterruptible power supply (UPS) for GLORIA-B are now in white, thermally conductive enclosures. Internal modules/circuit boards have been partially sealed, with thermal hotspots managed by copper strips and heat sink plates. The modified computing system is based on a 3U VPX-REDI single-board computer powered by a low-power Intel® Core<sup>TM</sup> i7 processor, which controls the GLORIA-B instrument through Gigabit Ethernet ports (Neubert et al., 2021). The storage capacity has been increased to 16 TB through the implementation of four solid state drives to enable flights with an extended duration of three to five days. The data storage is organized in a redundant RAID system. The real-time measurement data is streamed via an FPGA-based accelerator board directly to the storage system, where it can be efficiently downloaded through a 10 Gbit Ethernet downlink after each measurement flight. This facilitates the timely execution of post-processing operations.

# 2.1.5 Calibration system





The GLORIA instrument deployed on aircraft is equipped with two calibration blackbodies, one of them operated slightly below ambient temperature and one operated about 30 K above ambient temperature. Both of them can be heated or cooled and temperature stabilised using Peltier elements (Olschewski et al., 2013). Due to the restricted power budget on the balloon, a new blackbody with passive cooling had to be developed. The high flight altitude and an elevation angle of 40° for deep space measurements make it possible to derive the instrument self-emission directly from these measurements, therefore one calibration blackbody for gain determination is sufficient.

The pyramid structure of the backplane and the coating with Nextel Velvet Coating 811-21 have been inherited from GLORIA-AB, as well as the mounting of the temperature sensors inside the backplane. The same type of temperature sensors and the same temperature acquisition electronics as for GLORIA-AB have been used. The backplane measures 110 mm x 110 mm, and the cavity has a length of 130 mm. The blackbody is almost completely made of aluminium, and the parts are connected with closely spaced screws to provide good thermal contact.

The top side of the blackbody is exposed to deep space, while the bottom side is exposed to Earth (or the lower part of the gondola), and the outer side of the backplane is partly exposed

to Earth and partly to space. In order to obtain a homogeneous temperature across the backplane, the top and the backplane are painted with white Nextel Suede Coating 3101 428-04. In addition, a blank radiation shield made from stainless steel has been mounted below and behind the backplane to shield radiation from Earth and to reflect the radiation from the backplane into space (Fig. 1). The left and right side of the blackbody are kept with blank aluminium, and the bottom side has been insulated to avoid heating from Earth radiation. The blackbody temperature during flight ranged from about 240 K to 270 K, and although a small systematic gradient over the backplane can be observed, the overall homogeneity is still within  $\pm$  100 mK, which meets the requirement.

#### 2.1.6 Measurement scenario and automation







Since the vertical angular coverage of the instrument does not cover the whole altitude range of interest between flight altitude (about 36 km) and 5 km, two alternating elevation angles were chosen to cover this altitude range. This leads to an overlap of the vertical coverage of the two alternating measurements in the altitude range of about 26 to 29 km, varying with flight altitude.

Interferograms are Fourier transformed into the spectral domain and calibrated using in-flight measurements of a blackbody and deep space measurements. During the first flight a set of sixteen blackbody measurements and eight deep space measurements was taken about every 31 minutes for radiometric calibration. In the second flight in August 2022, an additional sequence of eight Nadir-looking measurements was included additionally in every measurement cycle.

An automated script ensures this sequence is followed without operator input. This script also ensures that the two alternating elevation angles are interchanged every four measurements. In specific measurement scenarios, the GLORIA-B system has the possibility to take over the command and control of the azimuthal orientation of the gondola by sending commands to an on-board interface provided by CNES. Such special measurement scenarios include constantly turning the gondola at a slow, but specific and constant rate while running the standard measurement sequence.

# 2.2 Generation of calibrated spectra

The level 1b processing from raw interferograms to calibrated spectra basically follows the procedure outlined by Kleinert et al. (2014) and Ungermann et al. (2022). Some important modifications are outlined below.

#### 275 Instrument offset determination:

In case of GLORIA-AB, deep space spectra are calibrated in a first step using the hot and the cold onboard blackbody. These pre-calibrated spectra are then used to perform a forward calculation of synthetic spectra representing the atmospheric contribution in the measurement. This contribution is then subtracted from the measurement in order to correct for the instrument offset (Ungermann et al., 2022). In case of GLORIA-B, only one onboard blackbody is flown, but deep space spectra are much less affected by atmospheric radiation due to the higher flight altitude. Therefore, it is possible to perform a pre-calibration using the blackbody and the deep space measurement with low spectral resolution, such that the atmospheric lines are strongly smoothed. The following fit and calculation of synthetic spectra is the same as for GLORIA-AB.

# Spectral calibration:





In case of GLORIA-AB, spectral calibration and the determination of the off-axis angle for each pixel is based on the spectral position of the CO<sub>2</sub> laser lines around 950 cm<sup>-1</sup> using deep space measurements. In case of GLORIA-B, the higher flight altitude and the larger elevation angle leads to much weaker line intensities in the spectra, such that the quality of the spectral calibration suffers from the low signal to noise ratio. Therefore, nominal atmospheric measurements taken with the upper elevation angle are used for this purpose. This proved to work well, because line intensities are sufficiently strong and even the lower pixel rows are not affected by clouds.

#### 295 Pixel binning and co-addition of spectra:

In case of GLORIA-AB, the spectra of all pixels of one row are averaged after filtering of bad pixels, such that the final level 1b product consists of one set of 128 vertical pixels. The same binning is done for the spectra of GLORIA-B, but in addition, eight consecutive measurements (four each with the upper and lower elevation angle, respectively) are combined to one dataset. In order to obtain the combined dataset on a regular angular grid, the grid of the measurements that were taken lower down is extended vertically, and the spectra of the measurements that

were taken higher up are interpolated onto this grid, using linear interpolation in the vertical direction. In the overlapping range, a linear weighting factor is introduced, such that a smooth transition is established between the two datasets. For the binned and combined datasets, the noise equivalent spectral radiance (NESR) finally amounts to about 5 x  $10^{-9}$  W/(cm<sup>2</sup> sr cm<sup>-1</sup>). In the overlapping range, the NESR is reduced by a factor of 0.7.

#### 2.3 Data analysis of measured spectra

In this study, we report results from the first GLORIA-B balloon flight carried out over northern Scandinavia on 21-22 August 2021 in the framework of the European Union Research Infrastructure HEMERA (Integrated access to balloon-borne platforms for innovative research and technology). Spectra were recorded with a maximum optical path difference of 8.0 cm which corresponds to a spectral sampling of 0.0625 cm<sup>-1</sup>. They were apodized with the Norton and Beer (1976) "strong" function. The high spectral resolution allows for the separation of individual spectral lines from continuum-like emissions such that many species even with minor emission contributions to the spectra can be retrieved with high spatial resolution. Flight path of the balloon gondola and the observed geolocations are displayed in Fig. 3. The gondola was launched from Esrange near Kiruna (67.9° N, 21.1° E). The float of up to 36 km lasted more than 12 hours and covered some time before sunset and after sunrise of the following day (between 17:11 UTC and 05:39 UTC). Due to weak stratospheric winds, the gondola did not move far away from the launch pad. Azimuth rotations were carried out continuously to probe the air masses around the gondola.

In order to analyse measured spectra, forward radiances are calculated with the Karlsruhe Optimized and Precise Radiative transfer Algorithm (KOPRA; Stiller et al., 2002) based on spectroscopic parameters from the HIgh-resolution TRANsmission molecular absorption database (HITRAN; Gordon et al., 2017). The procedure KOPRAFIT (Höpfner et al., 2002) uses derivatives (Jacobians) of the radiance spectrum with respect to atmospheric state and instrument parameters as calculated by KOPRA. The inverse problem is solved with a Gauss-Newton iterative method (Nocedal and Wright, 2006) in combination with a Tikhonov-Phillips regularization (Phillips, 1962; Tikhonov, 1963) using a constraint with respect to a first derivative of the a priori profile of the target species. The retrieval grid was set to 0.25 km up to 40 km. Above this altitude, the retrieval grid gradually gets coarser up to an altitude of 100 km.

Cloud-affected spectra were filtered out by a cloud index (in general larger than 2) taking into account the colour ratio of the mean radiance between 788.2-796.3 and 832.3-834.4 cm<sup>-1</sup> (Spang et al., 2004). In a further step, the pointing elevation angle was retrieved to check for a possible misalignment of the line of sight (LOS). The LOS error was estimated to 0.01° according to the method shown by Johansson et al. (2018). Prior to the trace gas analysis, the temperature retrieval was performed in spectral windows around 801, 811, 942, and 957 cm<sup>-1</sup> containing appropriate CO<sub>2</sub> transitions. The total error of the temperature retrieval was estimated to about 1.5 K (Johansson et al., 2018). Table 1 gives an overview of the spectral windows used for the GLORIA-B target gas retrieval performed in connection with this study. A careful selection of appropriate spectral windows is essential to perform retrievals with good accuracy. Main interfering species were fitted simultaneously together with the target gas. Minor species were either pre-fitted in a different spectral interval, or climatological profiles were used during the retrieval process. The error estimation of the retrieved molecules includes random noise, temperature errors, calibration (multiplicative gain), pointing, and field of view (FOV) inaccuracies, errors of non-simultaneously fitted interfering species, as well as spectroscopic data errors. All errors refer to the 1- $\sigma$  confidence limit.




Figure 3. Flight path of the balloon gondola on 21-22 August 2021 (black line) and GLORIA-B tangent

points (altitude colour coded). Measurement times are given in UTC. European Centre for Medium-Range Weather Forecasts (ECMWF) 100 hPa temperature field at 00 UTC is indicated in the background.

**Table 1.** Overview of GLORIA-B spectral windows for atmospheric species used in this study together with altitude resolution (alt. res.), typical noise errors and total errors, valid for a single vertical profile.

| Target<br>molecule | Spectral range<br>(cm <sup>-1</sup> )                                                                                                                                                      | Fitted species                                                                                                                                                                                           | Alt. res. (km) | Noise error (%) | Total error |
|--------------------|--------------------------------------------------------------------------------------------------------------------------------------------------------------------------------------------|----------------------------------------------------------------------------------------------------------------------------------------------------------------------------------------------------------|----------------|-----------------|-------------|
| O <sub>3</sub>     | 763.50 - 764.56<br>766.63 - 767.25<br>780.63 - 781.69<br>782.19 - 783.50<br>787.00 - 787.63<br>788.44 - 789.56<br>806.81 - 807.50<br>822.06 - 823.06<br>964.88 - 965.94<br>968.06 - 968.94 | $O_3$                                                                                                                                                                                                    | < 2            | 1 – 5           | 8 – 10      |
| CH <sub>4</sub>    | 1215.81 - 1217.00<br>1227.81 - 1228.50<br>1231.13 - 1232.56<br>1234.00 - 1235.25                                                                                                           | N <sub>2</sub> O, CH <sub>4</sub>                                                                                                                                                                        | < 2            | 2-5             | 20 – 22*    |
| CFC-11             | 840.50 – 859.69                                                                                                                                                                            | H <sub>2</sub> O, O <sub>3</sub> , HNO <sub>3</sub> , COCl <sub>2</sub> ,<br>CFC-11                                                                                                                      | < 2            | 1 – 5           | 5 – 15      |
| CFC-12             | 918.63 – 923.50                                                                                                                                                                            | H <sub>2</sub> O, HNO <sub>3</sub> , CFC-12                                                                                                                                                              | < 2            | 1 - 5           | 7 – 15      |
| HCFC-22            | 803.50 - 804.75<br>808.25 - 809.38<br>820.50 - 821.13<br>828.00 - 829.50                                                                                                                   | CO <sub>2</sub> , O <sub>3</sub> , HCFC-22                                                                                                                                                               | < 2            | 1 – 10          | 6 – 20      |
| CFC-113            | 801.00 - 819.50<br>819.50 - 830.00                                                                                                                                                         | H <sub>2</sub> O, O <sub>3</sub> , C <sub>2</sub> H <sub>6</sub> , HCFC-22,<br>CFC-113, CIONO <sub>2</sub>                                                                                               | < 4            | 3 – 20          | 7 – 30      |
| SF <sub>6</sub>    | 947.00 - 948.75                                                                                                                                                                            | $H_2O$ , $O_3$ , $COF_2$ , $SF_6$                                                                                                                                                                        | < 3            | 2 - 6           | 5 - 8       |
| HNO <sub>3</sub>   | 862.00 - 863.44<br>866.13 - 867.44<br>901.31 - 901.81                                                                                                                                      | $HNO_3$                                                                                                                                                                                                  | < 2            | 1 – 5           | 15 – 20*    |
| ClONO <sub>2</sub> | 779.88 - 781.00                                                                                                                                                                            | O <sub>3</sub> , ClONO <sub>2</sub>                                                                                                                                                                      | < 2            | 3 - 8           | 4 - 9       |
| $N_2O_5$           | 1227.25 – 1241.38                                                                                                                                                                          | H <sub>2</sub> O, O <sub>3</sub> , N <sub>2</sub> O, CH <sub>4</sub> ,<br>HOCl, COF <sub>2</sub> , N <sub>2</sub> O <sub>5</sub>                                                                         | < 2            | 2 – 4           | 5 – 8       |
| $NO_2$             | 801.00 - 820.00                                                                                                                                                                            | H <sub>2</sub> O, O <sub>3</sub> , NO <sub>2</sub> , HNO <sub>3</sub> ,<br>COF <sub>2</sub> , HCFC-22, CCl <sub>4</sub> ,<br>CFC-113, CIONO <sub>2</sub> ,<br>HNO <sub>4</sub> , PAN, BrONO <sub>2</sub> | < 5            | 2-5             | 5 – 15      |
| HNO4               | 801.00 - 820.00                                                                                                                                                                            | H <sub>2</sub> O, O <sub>3</sub> , NO <sub>2</sub> , HNO <sub>3</sub> ,<br>COF <sub>2</sub> , HCFC-22, CCl <sub>4</sub> ,<br>CFC-113, CIONO <sub>2</sub> ,<br>HNO <sub>4</sub> , PAN, BrONO <sub>2</sub> | < 4            | 3 – 8           | 5 – 25      |
| BrONO <sub>2</sub> | 801.00 - 820.00                                                                                                                                                                            | H <sub>2</sub> O, O <sub>3</sub> , NO <sub>2</sub> , HNO <sub>3</sub> ,<br>COF <sub>2</sub> , HCFC-22, CCl <sub>4</sub> ,<br>CFC-113, ClONO <sub>2</sub> ,<br>HNO <sub>4</sub> , PAN, BrONO <sub>2</sub> | < 30           | 9 – 15          | 10 – 30     |
|                    |                                                                                                                                                                                            | · · · · · · · · · · · · · · · · · · ·                                                                                                                                                                    |                |                 |             |

| PAN | 780.00 - 790.00 | $H_2O$ , $O_3$ , $HCFC-22$ , $CCl_4$ , | 5 15   | 10 - 30 |
|-----|-----------------|----------------------------------------|--------|---------|
|     | 794.00 - 805.00 | PAN                                    | 5 – 15 |         |

<sup>\*</sup>Large spectroscopic error.

#### 2.4 Model simulations





A multi-year simulation with the chemistry climate model ECHAM5/MESSy Atmospheric Chemistry (EMAC) was performed. This Eulerian model includes a number of submodels describing processes in the troposphere and middle atmosphere (Jöckel et al., 2010). The core model is the fifth generation European Centre Hamburg general circulation model (ECHAM5; Roeckner et al., 2006) that is connected to the submodels using the interface Modular Earth Submodel System (MESSy). For this study, an EMAC (MESSy version 2.55.0) multi-year simulation from 1979 until 2022 was used with a time step of 10 minutes and a spherical truncation of T42L90 (corresponding to a horizontal resolution of approximately 2.8° by 2.8° in latitude and longitude) with 90 hybrid pressure levels from the ground up to 0.01 hPa. The model was nudged towards the ECMWF ERA5 reanalysis data (Hersbach et al., 2020) to simulate realistic synoptic conditions. A comprehensive chemistry setup is included from the upper troposphere to the lower mesosphere. Rate constants of gas-phase reactions originate from Atkinson et al. (2007), Sander et al. (2011), and Burkholder et al. (2015). Photochemical reactions of short-lived bromine containing organic compounds are integrated into the model set-up (Jöckel et al., 2016) together with surface emissions of these species taken from scenario 5 of Warwick et al. (2006). The boundary condition of the greenhouse gases (including nitrogen oxides) was chosen according to the Coupled Model Intercomparison Project Phase 6 (CMIP6; Eyring et al., 2016). The model output was interpolated in time and space to the GLORIA-B measurement geolocations during the first flight.

#### 3 Results

#### 3.1 GLORIA-B measurements and external data comparison

In this section mean vertical profiles of O<sub>3</sub> and longer-lived trace gases retrieved from GLORIA-B measurements are compared to coincident in-situ ozonesonde, AirCore (Engel et al., 2017; Membrive et al., 2017; Laube et al., 2020), and cryosampler (Engel et al., 2016; Schuck et al., 2025) instruments. An AirCore is a passive sampling device based on a long coil of extremely thin-walled stainless-steel tubing (Karion et al., 2010). The newly developed

MegaAirCore consists largely of two connected pieces of coiled tubing (Laube et al., 2025). In contrast, the well-established whole air cryosampler holds 15 stainless steel sample flasks which were evacuated before the launch and opened and closed interactively during flight through telemetry commands from ground (Schuck et al., 2025). For CH<sub>4</sub> we also compared GLORIA-B data to SPECIES (SPECtromètre Infrarouge à lasErs in Situ) in-situ observations (Catoire et al., 2023). In the case of  $O_3$ , an additional comparison to Microwave Limb Sounder (MLS; e.g. Livesey et al., 2008; Hubert et al., 2016) satellite observations was performed. For this intercomparison, all GLORIA-B data points were averaged on a common altitude grid. This has the advantage that the noise part of the retrieval error is reduced by a factor  $1/n^{0.5}$  where n is the number of averaged data points. To determine whether a bias between the instruments is significant or not, combined errors which are calculated as the square root of the sum of the squares of the errors of each measurement are calculated according to Wetzel et al. (2022). To allow for a more realistic comparison in terms of altitude resolution, the data of the in-situ measurements were smoothed with the averaging kernel of the GLORIA-B retrieval.

**Figure 4.** Top: Azimuth orientation of the horizontal line of sight of the GLORIA-B instrument (0° = north,  $\pm 180^\circ$  = south). Bottom: Temporal evolution of ozone volume mixing ratio (ppmv) as seen from GLORIA-B over northern Scandinavia. The grey solid line shows the observer altitude ( $z_{obs}$ ). Black solid lines mark sunset and sunrise terminators. Dynamical tropopause (2 and 4 potential vorticity units from ECMWF) is plotted as dashed magenta lines. Changes in the ozone VMR around 17:18 UTC,

22:05 UTC, 23:13 UTC, and 05:02 UTC correlate with azimuthal rotations which indicate a gradient in the horizontal ozone amount.

### $3.1.1 O_3$





The monitoring of the expected recovery of the stratospheric ozone layer is still of scientific interest (Chipperfield and Bekki, 2024; WMO, 2022). Continuous ozone measurements are also important because this molecule is known to be an effective greenhouse gas in the upper troposphere and lowermost stratosphere (UTLS) region since it affects the radiation budget in this altitude range (Forster and Shine, 1997; Gettelman et al., 2011; Hansen et al., 1997; Riese et al., 2012; Xie et al., 2008). Figure 4 shows the ozone measurements performed with GLORIA-B between 17:11 UTC on 21 August 2021 and 05:39 UTC on the following day during float of the gondola. The vertical ozone distribution reflects the expected increase of ozone VMR with altitude reaching maximum values of about 6 ppmv near 35 km. Short-term variations in the ozone mixing ratio at 17:18 UTC, 22:05 UTC, 23:13 UTC, and 05:02 UTC are due to the changes in the azimuthal viewing direction and reflect the horizontal variation in the amount of ozone.

Figure 5. Left: All measured GLORIA-B  $O_3$  data points (orange squares) and mean vertical profile (red solid line) compared to high-resolution ozonesonde data (green squares) and these data smoothed with the averaging kernel of GLORIA-B (solid dark green line, dotted outside altitude region of reliability). Satellite measurements (version 5) from the Microwave Limb Sounder (MLS), averaged in  $\pm 5$  deg. latitude and  $\pm 20$  deg. longitude bins around the GLORIA-B mean observation are shown, too. Middle

and right: Absolute and relative differences together with combined error bars and GLORIA-B standard deviation (SD).

Figure 5 shows the mean ozone profile derived from the measured GLORIA-B spectra. As previously mentioned, the gondola did not travel far horizontally from the launch pad, maintaining a close horizontal distance of less than 100 km to the ozonesonde. This proximity, along with the circular rotating line of sight of GLORIA-B, allows for a direct comparison between both measurements. The ozonesonde data were smoothed using the averaging kernel of GLORIA-B. Above 30 km, the ozonesonde values gradually become unrealistic, a typical behavior of ozonesondes (Stauffer et al., 2022). VMR differences between the mean GLORIA-B ozone profile, the smoothed ozonesonde, and MLS remain within 10% for most altitudes. A slight positive bias in the GLORIA-B profile is observed above 22 km (compared to the ozonesonde) and between 20 and 32 km (compared to MLS). However, these biases are not significant when considering the combined error bars of each measurement.

**Figure 6.** Same as Fig. 4 but for CH<sub>4</sub>.

#### 3.1.2 CH<sub>4</sub>




Methane is an important greenhouse gas and further increasing concentrations are expected during this century (WMO, 2022). Furthermore, the oxidation of CH<sub>4</sub> in the stratosphere

produces two H<sub>2</sub>O molecules per CH<sub>4</sub> molecule, making CH<sub>4</sub> an important source of stratospheric H<sub>2</sub>O (e.g. Brasseur and Solomon, 2005).





The CH<sub>4</sub> cross section as measured by GLORIA-B is given in Fig. 6. The CH<sub>4</sub> VMR exhibits the expected behavior with high tropospheric values close to 1.9 ppmv and decreasing amounts in the stratosphere. The overall structure appears a bit noisy. CH<sub>4</sub> variations due to changes in the azimuth are barely visible since the horizontal gradient of CH<sub>4</sub> concentration is small.

**Figure 7.** Left: All measured GLORIA-B CH<sub>4</sub> data points (orange squares) and mean vertical profile (red solid line) compared to in-situ AirCore (cyan and pink squares) and SPECIES data (green squares) and these data smoothed with the averaging kernel of GLORIA-B (solid blue, violet and dark green line). For comparison, a northern hemispheric lower tropospheric value from the AGAGE network is shown, too (magenta star). Middle and right: Absolute and relative differences together with combined error bars and GLORIA-B standard deviation (SD).

The comparison of the mean GLORIA-B CH<sub>4</sub> profile to in-situ measurements is displayed in Fig. 7. The overall shape of the vertical profiles of all observations is consistent. The difference between GLORIA-B and the in-situ measurements is within 10 %. However, there is a small positive bias visible in the GLORIA-B data that is not significant since the deviations are clearly within the combined error bars. Note that the CH<sub>4</sub> uncertainty in the GLORIA-B observation is dominated by the large spectroscopic error (mainly line strength and line width) given in the HITRAN database (Gordon et al., 2017). It seems that the spectroscopic error is overestimated, since the difference between GLORIA-B and the in-situ instruments is clearly smaller than the

combined error itself. All measurements at 10 km are close to the northern hemispheric lower tropospheric value from the Advanced Global Atmospheric Gases Experiment (AGAGE) network (Prinn et al., 2018).

#### 470 3.1.3 CFC-11




The molecule CFC-11 (CCl<sub>3</sub>F) is a rather long-lived chlorofluorocarbon that is removed from the atmosphere mainly by stratospheric photolysis (e.g. Ko and Dak Sze, 1982). Since it belongs to the main ozone depleting substances it is necessary to monitor the concentration of this trace gas to detect unexpected emissions (WMO, 2022).

**Figure 8.** Left: All measured GLORIA-B CFC-11 data points (orange squares) and mean vertical profile (red solid line) compared to in-situ MegaAirCore (cyan squares) and cryosampler data (green squares) and these data smoothed with the averaging kernel of GLORIA-B (solid blue and dark green line). For comparison, a northern hemispheric lower tropospheric value from the AGAGE network is shown, too (magenta star). Middle and right: Absolute and relative differences together with combined error bars and GLORIA-B standard deviation (SD).

Figure 8 presents the mean retrieved GLORIA-B CFC-11 profile in comparison to the in-situ MegaAirCore and cryosampler observations. The rapid decrease in volume mixing ratio (VMR) with altitude due to photolysis is evident in all measured CFC-11 profiles, approaching values close to zero above approximately 26 km. Below 17 km, measured CFC-11 values of all three instruments lie close together (within 10 %) and inside the combined error limits. Above this

altitude we see an agreement between GLORIA-B and the cryosampler data with respect to the combined errors but a significant positive bias of the MegaAirCore observation. This bias is caused by a combination of internal MegaAirCore air mixing and averaging of air masses due to refilling into subsampling canisters (Laube et al., 2025). It is therefore especially pronounced for species with a large concentration gradient between 19 and 24 km. At an altitude of 10 km, all measurements align closely with the northern hemispheric lower tropospheric AGAGE value.

**Figure 9.** Same as Fig. 8 but for CFC-12.

#### 3.1.4 CFC-12





The chlorofluorocarbon CFC-12 (CCl<sub>2</sub>F<sub>2</sub>) is, like CFC-11, removed from the stratosphere mainly by photolysis (e.g. Ko and Dak Sze, 1982). Since the atmospheric lifetime of CFC-12 (around 102 years) is about twice as long as for CFC-11 (Hoffmann et al., 2014), the stratospheric decrease of the CFC-12 VMR with altitude is less pronounced than in the case of CFC-11. Figure 9 illustrates the comparison of vertical CFC-12 profiles observed by GLORIA-B and the in-situ instruments. Due to technical problems, the MegaAirCore device collected measurements only within a limited altitude range between 23 and 26 km which are in line with the GLORIA-B observations. The cryosampler data show agreement (within 10 %) with the mean GLORIA-B profile taking into account the combined error limits. At an altitude

of 10 km, the GLORIA-B data points are close to the northern hemispheric lower tropospheric AGAGE value.

#### 510 3.1.5 HCFC-22




The longer-lived hydrofluorocarbon HCFC-22 (CHClF<sub>2</sub>) is often used as an alternative to the highly ozone-depleting substances CFC-11 and CFC-12 such that its tropospheric concentration increased until the early 2020s (e.g. Chirkov et al., 2016; WMO, 2022). The comparison between GLORIA-B and the in-situ observations is shown in Fig. 10. The difference between both in-situ instruments and the retrieved GLORIA-B data is less than 10 % below about 19 km. However, above this altitude, a clear negative bias is evident in the cryosampler VMR, which becomes significant between 19.5 and 23 km. A similar negative bias has already been observed in cryosampler data comparisons with ACE-FTS (Atmospheric Chemistry Experiment - Fourier Transform Spectrometer) satellite measurements (Kolonjari et al., 2024). Although the reason for this bias is still unclear there is an indication that the O<sub>3</sub> present in the cryosamples might affect species like HCFC-22 chemically during the sampling or storage process (Laube et al., 2025). In contrast, a positive bias is evident in the MegaAirCore data up to 24 km altitude. This positive bias is caused by the same problems as described in section 3.1.3.

Figure 10. Same as Fig. 8 but for HCFC-22.

#### 3.1.6 CFC-113





The chlorofluorocarbon CFC-113 (CCl<sub>2</sub>FCClF<sub>2</sub>) is also very unreactive in the troposphere and is photolyzed in the stratosphere (e.g. Sen et al., 1999; Dufour et al., 2005). Due to the significant spectral overlap from many interfering species, retrieving the VMR of CFC-113 is more challenging compared to the previously discussed target gases. Figure 11 illustrates the intercomparison of vertical profiles from GLORIA-B and the in-situ instruments. Up to 17 km altitude, the VMR differences are less than 10 % and fall within the combined error limits. However, above this altitude, MegaAirCore values are significantly larger than those retrieved by GLORIA-B. In this upper altitude region, the cryosampler values are mostly somewhere between GLORIA-B and MegaAirCore. It should be noted that the standard deviation of the mean GLORIA-B profile is large, as the CFC-113 signal in the GLORIA-B spectra is weak at higher altitudes.

Figure 11. Same as Fig. 8 but for CFC-113.

# 3.1.7 SF<sub>6</sub>

The long-lived anthropogenic tropospheric source gas sulfur hexafluoride (SF<sub>6</sub>) is a trace gas with a high global warming potential (Hu et al., 2023). Since this molecule is chemically inert in the troposphere with a monotonic increase in atmospheric mixing ratio, it is suitable to study tropospheric and stratospheric transport processes based on the concept of age of air determination (e.g. Hall and Waugh, 1998; Engel et al., 2009; Stiller et al., 2012; Ray et al.,

2014; Schuck et al., 2024; Saunders et al., 2025; Voet et al., 2025). The comparison of GLORIA-B retrieved SF<sub>6</sub> and the in-situ instruments is displayed in Fig. 12. The decreasing SF<sub>6</sub> values from 10 to about 25 km reflect the increasing age of air in the upper altitude range compared to the lowermost stratosphere (near 10 km). Differences between all measurements are less than 10 % and stay within the combined error bars. The VMR peak seen in the GLORIA-B data around 29 km appears not to be significant since the standard deviation is large in this altitude region. At an altitude of 10 km, the GLORIA-B data points correspond to the northern hemispheric lower tropospheric AGAGE value.

**Figure 12.** Same as Fig. 8 but for SF<sub>6</sub>.





#### 3.2 GLORIA-B measurements and model comparison

In this section we demonstrate that GLORIA-B is able to measure temporal changes in the mixing ratios of photochemically active nitrogen (N<sub>2</sub>O<sub>5</sub>, NO<sub>2</sub>), chlorine (ClONO<sub>2</sub>), and bromine (BrONO<sub>2</sub>) containing compounds with high spatial and temporal resolution. Since no external measurements are available for comparison, we compare GLORIA-B data with EMAC chemical model simulations as part of a consistency check. Previous comparisons with MIPAS balloon observations have shown that EMAC can reliably simulate daily cycles of these molecules (Wetzel et al., 2012, 2017).

# 3.2.1 ClONO<sub>2</sub>

Chlorine nitrate (ClONO<sub>2</sub>) is an important compound in stratospheric chemistry, particularly in the context of ozone layer depletion (e.g. Brasseur and Solomon, 2005; von Clarmann and Johansson, 2018). Since it serves as a key reservoir species for reactive chlorine in the stratosphere, it temporarily stores chlorine atoms that could otherwise participate in ozone-depleting reactions. The diurnal variation of ClONO<sub>2</sub> as observed by GLORIA-B is shown in Fig. 13a. ClONO<sub>2</sub> undergoes minor diurnal variations at higher altitudes as a result of the interaction between the ClONO<sub>2</sub> build-up via NO<sub>2</sub> and ClO and the ClONO<sub>2</sub> decay by photolysis, resulting in somewhat higher values during night compared to day. Changes in the VMR around 17:18 UTC, 22:05 UTC, 23:13 UTC, and 05:02 UTC correlate with azimuthal rotations which indicate a gradient in the horizontal distribution of the species. The EMAC simulation reproduces the observation (Fig. 13b) such that differences between GLORIA-B and EMAC are not larger than ±0.3 ppbv (Fig. 13c).

**Figure 13.** Temporal evolution of ClONO<sub>2</sub> as seen by GLORIA-B on 21-22 Aug. 2021 above northern Scandinavia (**a**). The variation of the azimuthal viewing direction is given on top (red dots). The observer altitude is shown as grey line. Black solid lines mark the sunset (SS) and sunrise (SR) terminators. The dynamical tropopause using 2 and 4 potential vorticity (PV) units is given as dashed magenta lines. The corresponding simulation by EMAC (**b**) and the difference (**c**) between the measurement and the chemical model are also shown.

**Figure 14.** Same as Fig. 13 but for  $N_2O_5$ . A pronounced diurnal variation of the amount of  $N_2O_5$  is evident both in the measured GLORIA-B data and in the EMAC simulation. The strong correlation between azimuthal line of sight rotations and the changing  $N_2O_5$  VMR due to different times of illumination is clearly visible.

# $3.2.2 N_2O_5$

Dinitrogenpentoxide (N<sub>2</sub>O<sub>5</sub>) is, besides HNO<sub>3</sub> and ClONO<sub>2</sub>, a stratospheric reservoir species for NO<sub>2</sub> and NO (e.g. Brasseur and Solomon, 2005). It is produced during the night from reactions with O<sub>3</sub>, NO<sub>2</sub> and NO<sub>3</sub>. After sunrise, N<sub>2</sub>O<sub>5</sub> gradually photolyzes back into NO<sub>2</sub> and

NO<sub>3</sub>. Therefore, the amount of N<sub>2</sub>O<sub>5</sub> shows a clear daily cycle in the stratosphere (Wiegele et al., 2009; Wetzel et al., 2012). A significant increase in the N<sub>2</sub>O<sub>5</sub> VMR after local sunset is visible in the GLORIA-B measurements (see Fig. 14a). Azimuthal rotations of the line of sight around 22:05 UTC and 23:13 UTC correlate clearly with the observed amount of N<sub>2</sub>O<sub>5</sub>. The EMAC simulation reproduces the measured N<sub>2</sub>O<sub>5</sub> VMR close to perfect (Fig. 14b). Consequently, deviations between measurement and model are small despite minor systematic differences (Fig. 14c).

**Figure 15.** Same as Fig. 13 but for NO<sub>2</sub>. The measured data (a) have been temporally smoothed with a 17-point adjacent averaging routine to attenuate noisy structures. Therefore, no changes in the measured VMR due to azimuthal line of sight rotations are visible.

# 3.2.3 NO<sub>2</sub>






Nitrogen dioxide (NO<sub>2</sub>) is not only involved in the temporal cycle of N<sub>2</sub>O<sub>5</sub> but also shows a strong interaction with NO (e.g. Brasseur and Solomon, 2005). This results in a pronounced diurnal variation of the NO<sub>2</sub> VMR (Wiegele et al., 2009; Wetzel et al., 2012). This is also visible in the GLORIA-B measurements (Fig. 15a) and the EMAC simulation (Fig. 15b). The start of the NO<sub>2</sub> VMR build-up at 34 km in the measurement is reproduced slightly delayed in the model. The small secondary maximum near 22 km (after sunrise) seen in the measurement is not visible in the model calculation. However, it should be emphasized that in this part of the measurement the retrieval noise error is enhanced such that this secondary maximum may also be attributed to a retrieval artefact. In general, the NO<sub>2</sub> values from GLORIA-B are slightly higher in most regions as simulated with EMAC. All VMR differences between GLORIA-B and EMAC are displayed in Fig. 15c.

# 3.2.4 BrONO<sub>2</sub>

Bromine nitrate (BrONO<sub>2</sub>) is the most abundant inorganic bromine (Br<sub>y</sub>) compound in the stratosphere (e.g. Brasseur and Solomon, 2005). It is produced from BrO and NO<sub>2</sub> during night and it is destroyed during the day by photolysis and reaction with atomic oxygen which results in a pronounced diurnal cycle of BrONO<sub>2</sub>. The first stratospheric measurements of BrONO<sub>2</sub> were reported by Höpfner et al. (2009) who derived vertical profiles of BrONO<sub>2</sub> from atmospheric infrared emission spectra recorded by MIPAS aboard Envisat. The first observations of stratospheric diurnal variations of this substance were enabled by the analysis of spectra measured by MIPAS-B, the balloon version of MIPAS (Wetzel et al., 2017). The temporal evolution of BrONO<sub>2</sub> as seen by GLORIA-B is depicted in Fig. 16a. The nighttime increase of the amount of BrONO<sub>2</sub> and its daytime decay (towards BrO) are clearly visible. This variation is largely reproduced by EMAC (Fig. 16b). Differences between measurement and model above about 16 km (Fig. 16c) are obvious in the absolute amount of available bromine in the model that is dependent on the surface emission scenario of short-lived bromine-containing organic compounds (Warwick et al., 2006). Differences below this altitude are difficult to interpret since GLORIA-B errors there exceed the 100 % limit.

**Figure 16.** Same as Fig. 13 but for BrONO<sub>2</sub>. The measured data (a) have been temporally smoothed with a 39-point adjacent averaging routine to attenuate noisy structures. Therefore, no changes in the measured VMR due to azimuthal line of sight rotations are visible. The cyan coloured box (a) marks the region for estimation of 'measured' Br<sub>y</sub>.

As mentioned, BrONO<sub>2</sub> is the dominant species of stratospheric inorganic bromine. In our case, simulations of EMAC indicate that more than 90 % of nocturnal (20:06 UTC - 01:21 UTC) Bry is in the form of BrONO<sub>2</sub> between 24 and 29 km. According to the method described by Wetzel et al. (2017), we can estimate 'measured' inorganic bromine [Bry(meas)] in this nocturnal altitude region by scaling measured [BrONO<sub>2</sub>(meas)] with the modeled ratio

 $[Br_v(mod)]/[BrONO_2(mod)]$  from EMAC. We then calculate mean  $[Br_v(meas)]$  to 20.4  $\pm$  2.5 635 pptv which is a realistic value. The resulting error bar represents the 1- $\sigma$  total error originating from [BrONO<sub>2</sub>(meas)]. Voss et al. (2024) recently determined stratospheric [Br<sub>v</sub>] of 17.5  $\pm$  2.2 pptv from the ratio [BrO] / [Br<sub>v</sub>] using differential optical absorption spectroscopy (DOAS) BrO observations from a balloon flight carried out from Timmins (Ontario, Canada) on 23 640 August 2022, in combination with chemical modelling. Although the Br<sub>v</sub> values estimated from measurements of the two different instruments differ slightly, there is still an overlap between 17.9 and 19.7 pptv taking into account the error bars. A similar positive bias of Br<sub>v</sub> deduced from mid-infrared BrONO<sub>2</sub> measurements in comparison to UV-visible Br<sub>y</sub> inferred from BrO observations was also recognized and discussed by Wetzel et al. (2017). Since molecular 645 spectroscopy plays an important role in the accuracy of mid-infrared and UV-visible (DOAS) measurements, the observed Bry difference might be explained at least partly by unquantified inaccuracies in the spectroscopic parameters (mainly line intensity).

#### 4 Conclusions





Vertical profiles of a number of trace gases were retrieved from limb-imaging infrared emission spectra measured during the first flight of the GLORIA balloon instrument above northern Scandinavia. First, long-lived tropospheric source gases and stratospheric tracer species were compared to external measurements to assess the quality and the reliability of the GLORIA-B observations. Note that the accuracy of the measured gases depends mainly on the strength of spectral signatures of the target gas and the separability of these signatures from those of interfering gases. For the gases O<sub>3</sub>, CH<sub>4</sub>, SF<sub>6</sub>, and CFC-12 we find an overall agreement within 10 % between GLORIA-B retrieved vertical profiles and the in-situ MegaAirCore and cryosampler instruments. No unexplained biases between GLORIA-B and the in-situ instruments are detectable, the differences stay within the combined error limits. Bearing in mind this agreement, the large estimated spectroscopic CH<sub>4</sub> error (see Table 1) originating from the HITRAN database appears to be overestimated. For the molecules CFC-11, HCFC-22, and CFC-113 we find a level of agreement within 10 % to 20 % up to about 18 km, and differences are inside the combined error limits. Above this altitude region, deviations between GLORIA-B and also between the MegaAirCore and cryosampler instruments become larger and at least partly significant with respect to the combined errors. However, a bias in the MegaAirCore altitude assignment due to diffusion cannot be ruled out. This could partly explain a bias between MegaAirCore and GLORIA-B, and also a bias between MegaAirCore and the cryosampler instrument, especially in the vicinity of strong concentration gradients.

Second, photochemically active species (N<sub>2</sub>O<sub>5</sub>, NO<sub>2</sub>, ClONO<sub>2</sub>, and BrONO<sub>2</sub>) are compared to simulations performed with the chemistry climate model EMAC. Calculations largely reproduce the temporal evolution of the observed gases while some differences appear in the absolute VMR values of the gases. While for N<sub>2</sub>O<sub>5</sub>, NO<sub>2</sub>, and ClONO<sub>2</sub> the absolute amount differs only slightly, the difference for BrONO<sub>2</sub> is significantly larger (up to 3 pptv in the region of the BrONO<sub>2</sub> VMR maximum). However, it should be emphasized that the absolute VMR of BrONO<sub>2</sub> in the model simulation is strongly dependent on the assumed available inorganic bromine, which was chosen according to scenario 5 by Warwick et al. (2006).

In conclusion, the first flight of the GLORIA balloon instrument demonstrated its capability to retrieve accurate vertical profiles of trace gases, showing agreement with external measurements (taking into account the combined error limits) over wide altitude ranges. The instrument also provides measurements of the temporal evolution of trace gases. Hence, GLORIA-B and, in general the limb-imaging technique, can contribute to a better understanding of the distribution and temporal evolution of key atmospheric species from the upper troposphere to the middle stratosphere, and supports climate relevant process understanding that allows continuous atmospheric model improvements.


Data availability. GLORIA-B measurements will be available on the KITopen repository. EMAC data are available upon request. MegaAirCore CH<sub>4</sub> and part of cryosampler data is published on the Zenodo repository.

Author contributions. GW wrote most parts of the manuscript and performed the bulk of the data analysis, with input from all co-authors. AK managed the instrumental part of the manuscript. SJ, AK, JU, MH, PP, and WW performed the GLORIA data processing and were involved in many helpful scientific discussions. TK, TN, HN, JS, HS, GS, and AS developed the necessary modifications to bring the GLORIA instrument to the balloon platform. FFV, EK,
 GM, HN, CP, TG, MRe, and TN, operated GLORIA during the balloon campaign in Kiruna. CC and JL provided MegaAirCore data. AE, TS, and JL analyzed cryosampler measurements.

VC and PJ provided SPECIES observations. OK performed the EMAC simulations. PB and MRi directed the research. All authors commented on and improved the manuscript.

Competing interests. One author is a member of the editorial board of AMT.

Acknowledgements. We thank the Centre National d'Études Spatiales (CNES) for the excellent balloon operations of the GLORIA-B instrument from Kiruna, Sweden. These operations were carried out under the auspices of the HEMERA research infrastructure. Results are based on the efforts of all members of the GLORIA team, including the technology institutes ZEA-1 and ZEA-2 at Forschungszentrum Jülich and the Institute for Data Processing and Electronics at the Karlsruhe Institute of Technology. This project has received support from the European Commission in the frame of the INFRAIA grant 730970-HEMERA. Support of ESA (projects MAGIC4AMPAC and CAIRTEX) is acknowledged. The EMAC simulations were performed on the supercomputer HoreKa at the National High-Performance Computing Center at KIT (NHR@KIT). This center is jointly supported by the Federal Ministry of Education and Research and the Ministry of Science, Research and the Arts of Baden-Württemberg, as part of the National High-Performance Computing (NHR) joint funding program. The development of the new blackbody was supported by the MetEOC-4 project through the European Metrology Programme for Innovation and Research (EMPIR) under Grant 19ENV07. The EMPIR programme is co-financed by the Participating States and from the European Union's Horizon 2020 research and innovation programme. We thank C. Monte, M. Reiniger and A. Adibekyan at PTB for their support in developing and characterising the blackbody.

Financial support. We acknowledge support by Deutsche Forschungsgemeinschaft and Open Access Publishing Fund of Karlsruhe Institute of Technology. Cryosampler analysis at Goethe University Frankfurt was funded through the DFG collaborative research program: The Tropopause Region in a Changing Atmosphere TRR 301 – Project-ID 428312742.

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
