# Peer review of "Intercomparison and validation of first GLORIA-B measurements of stratospheric and upper tropospheric long-lived tracers and photochemically active species"

_EGUsphere, 2025_

## Author Comment (AC1)

**Response to Referee 1:**

First of all, we thank the referee for the effort to carefully reading the manuscript and for all comments.

**General comments:**

*This paper describes the first result of GLORIA-B flight from Kiruna in 2021. The paper is generally well written and the contents which are described in the paper is clear. I felt the paper is almost worth published in Atmospheric Measurement Techniques. I have only a few minor points which would be nice to be modified before publication, which is pointed out below.*

**Minor comments:**

*1) P.1, L.31: What is "HEMERA"? Please provide what it means for.*

HEMERA is not an acronym and stands for: "Integrated access to balloon-borne platforms for innovative research and technology." HEMERA is a Research Infrastructure funded by the Horizon 2020 framework Programme of the European Union which integrates a large starting community in the field of tropospheric and stratospheric balloon-borne research, to make existing balloon facilities available to all scientific teams in the European Union, Canada and associated countries. That means that although the name HEMERA is not an official abbreviation, it was chosen as the project name, inspired by the Greek goddess Hemera, the personification of day. We integrated the meaning of HEMERA into the text of the manuscript.

*2) P.22, L.510: "a clear negative bias is evident in the cryosampler VMR." What is the cause of this negative bias? Please give some idea for this bias.*

A similar negative bias has already been observed in cryosampler data comparisons with ACE-FTS (Atmospheric Chemistry Experiment - Fourier Transform Spectrometer) satellite measurements (see Fig. 4 in Kolonjari et al., Atmos. Meas. Tech., 17, 2429–2449, https://doi.org/10.5194/amt-17-2429-2024, 2024). However, the reason for this bias is still unclear although there is an indication that the $O_3$ present in the cryosamples might affect some species (like HCFC-22) chemically during the sampling or storage process (see Fig. 4 in Laube et al., Atmospheric Meas. Tech., 18, 4087–4102, https://doi.org/10.5194/amt-18-4087-2025, 2025). We added this information to the manuscript text.

*3) P.23, L.539-540: What is the meaning of the sentence? "The vertical shape of the observed profiles is largely as expected." I guess some word(s) are missing.*

In fact, this sentence is not very specific. Therefore, we rewrote it to: "The decreasing $SF_6$ values from 10 to about 25 km reflect the increasing age of air in the upper altitude range compared to the lowermost stratosphere (near 10 km)."

---

## Author Comment (AC2)

**Response to Referee 2:**

First of all, we thank the referee for the effort to carefully reading the manuscript and for all comments.

**General comments:**

*GLORIA-B is the balloon adapted version of the GLORIA instrument. Wetzel et al. describe in their manuscript the GLORIA-B instrument and present the first measurements that have been made with GLORIA-B during the HEMERA campaigns in August 2021 and 2022. The during these campaigns derived GLORIA-B measurements are compared with in-situ measurements from e.g. MegaAirCore and with model simulations from EMAC. The comparisons presented in Wetzel et al. show that the observations from GLORIA-B are in good agreement with in-situ measurements and that the temporal evolution of the measured trace gases is in agreement with model simulations from EMAC.*

*The manuscript is well written and the presented results worth to be published. I have only some suggestions for minor revisions that I think will improve the manuscript.*

**Specific comments:**

*P1, Title: The title is too general and does not really reflect what is the main purpose of the study. To my understanding this is the first publication on GLORIA-B and thus the instrument is here presented and validated for the first time. This should be clearly reflected in the title.*

We rewrote the title to: "Intercomparison and validation of first GLORIA-B measurements of stratospheric and upper tropospheric long-lived tracers and photochemically active species" since the paper includes the validation aspect and the intercomparison to measurements of some independent instruments.

*P1, Abstract: Also, the abstract should be revised. The current version reads like a summary rather than an abstract. I would suggest to add a general sentence to motivate why such measurements are needed. Also add some sentences on what has been done and what the main purpose of this study is (characterization and validation of GLORIA-B). Then you can provide a short summary of the major findings (good agreement with other measurement and model simulation) and then conclude what benefit the scientific community has from such kind of measurements.*

We rewrote the abstract slightly and added sentences according to the referee's suggestions.

*P1, L31: The abbreviation HEMERA needs to be introduced.*

HEMERA is not an acronym and stands for: "Integrated access to balloon-borne platforms for innovative research and technology." HEMERA is a Research Infrastructure funded by the Horizon 2020 framework Programme of the European Union which integrates a large starting community in the field of tropospheric and stratospheric balloon-borne research, to make existing balloon facilities available to all scientific teams in the European Union, Canada and associated countries. That means that although the name HEMERA is not an official

abbreviation, it was chosen as the project name, inspired by the Greek goddess Hemera, the personification of day. We integrated the meaning of HEMERA into the text of the manuscript

*P3, L71: "some time" is a bit vague. Can you provide some numbers?*

MIPAS needed about a minute for one vertical scan (vertical profile). This duration depended slightly on the observation mode used and the phase of the mission (before or after 2004, when the reduced spectral resolution mode was introduced), but the order of magnitude remains at about one minute per vertical profile. We added this information to the text.

*P3, L80: From this statement I understand that this is the first publication on GLORIA-B. If that is the case such a statement should already appear in the abstract.*

This aspect is now presented more clearly in the abstract.

*P3, L80-85: Actually, that paragraph I would have rather expected in the abstract than in the introduction. It is fine to have it in both, but definitely some sentences like these should be found in the abstract. In general, your abstract provides all necessary information, but the message gets not that easily and clearly through as it is the case in this paragraph.*

This aspect is now also presented more clearly in the abstract.

*P12, L326: This altitude? Which altitude exactly? 40 km? Add the altitude so that it is clear which altitude is meant.*

Above 40 km, the retrieval grid gradually gets coarser up to an altitude of 100 km. We added this information to the text.

*P14, Numbers: For the spectral ranges, I would suggest to use only two digits after the comma for better readability.*

Agreed. We reduced the number of digits to two.

*P15, L372: Add a header "Results" to be more clear that from here on the description of the results starts?*

We included a header "Results" and adjusted the corresponding numbering of the following sections in the manuscript.

*P24, L549 and P25, L564: On P24 you write that you will demonstrate that GLORIA is able to measure the temporal evolution of trace gases by comparing to model simulations. On P25 however you write that EMAC is able to reproduce the observations. Are you now evaluating the model or the observations? Check that you have a consistent way of writing/discussing the results in this section.*

Normally a model is validated by measurements but in this case, we have a new instrument and therefore it is not self-evident that it measures all parameters reliably. Since no external measurements are available for comparison, we compare GLORIA-B data with EMAC chemical model simulations as part of a consistency check. Previous comparisons with MIPAS

balloon observations have shown that EMAC can reliably simulate daily cycles of these molecules. We changed the corresponding text and mention two MIPAS-B related references (Wetzel et al., 2012, 2017) which are already cited in the manuscript.

*P29, Figure 16: I would suggest to use another colour than cyan since it is hardly visible. Use a somewhat darker colour.*

We enlarged the thickness of the cyan coloured box such that it is clearly recognizable now.

*P29, L617: Here a statement or a reference if these are realistic results or not is missing.*

More than 90 % of nocturnal inorganic bromine in EMAC is in the form of $BrONO_2$ between 24 and 29 km. This can be directly deduced from the bromine partitioning of EMAC by considering the $BrONO_2$/$Br_y$ ratio. The $Br_y$ value obtained is realistic. This method is described in the manuscript text and shown in more detail in Wetzel et al. (2017). This reference is already given in the text.

*P30, L650ff: In the conclusions you clearly write that there are also differences and areas where the results are not that perfect. When reading the result section I had the feeling that the results are solely good. Check for consistency and also mention in the respective result subsections where and when deviations occur.*

We modified the corresponding text in the "Results" and "Conclusions" sections to make these statements consistent.

*P31, L658: "significantly larger" -> please quantify.*

The difference is up to 3 pptv in the region of the $BrONO_2$ VMR maximum. We added this information to the text.

*P31, L668: This statement is a bit contradicting to what has said before. Of course, with your measurements you can improve models, but your motivation for showing comparisons to model simulations was to show that GLORIA-B can measure the temporal evolution of the trace gases.*

Yes, we aim to demonstrate both that diurnal cycles of trace gases can be measured with GLORIA-B and that these measurements can help improve atmospheric models (in this case the first GLORIA-B measurements were also used as part of a consistency check with the EMAC model). We rephrased the text (section 3.2 and conclusions) to make this aspect clearer.

**Technical corrections:**

*P3, L67: Add "the" so that it reads the "Global Ozone Monitoring……"?*

We added "the" to the corresponding text.

*P6, L141: Remove colon after subsection title.*

Okay.

*P7, L154: Same here as for P6, L141.*

Okay.

*P9, L239: or the lower part of the gondola -> or "the" lower part of the gondola?*

The lower part of the gondola. We modified the manuscript text accordingly.

*P12, L304: 21/22 August -> 21-22 August 2021*

Okay.

*P12, L319: Use also capital letters for the written out version of HITRAN? For KOPRA this is done.*

Okay.

*P12, L328: Are the digits after the comma needed? For better readability I would suggest to omit these.*

Since the microwindows are not large we think we should give at least one digit after the comma so we changed this in the text accordingly.

*P13, L344: 21/22 -> 21-22 (this should also be corrected in the figures)*

We changed this in all corresponding places in the text and in the figures.

*P14, Table 1 caption: alt. reso. In parenthesis obsolete. This is also without writing this clear. However, I would in the table header abbreviate resolution with res. Instead of reso.*

Okay, we changed "reso" to "res".

*Figure 7 and other figures of this kind: On a printed version the squares in the legend are hardly visible and in the figure itself I can see them neither in the printed version nor in the pdf version.*

We enlarged the squares in the corresponding figures to make them better visible.

*P31, L672: Add "published" so that it reads "is published on Zenodo".*

Okay, we changed this in the text.